# Netherton Syndrome Caused by Heterozygous Frameshift Mutation Combined with Homozygous c.1258A>G Polymorphism in *SPINK5* Gene

**DOI:** 10.3390/genes14051080

**Published:** 2023-05-14

**Authors:** Chiara Moltrasio, Maurizio Romagnuolo, Davide Riva, Davide Colavito, Silvia Mariel Ferrucci, Angelo Valerio Marzano, Gianluca Tadini, Michela Brena

**Affiliations:** 1Dermatology Unit, Fondazione IRCCS Ca’ Granda Ospedale Maggiore Policlinico, 20122 Milan, Italy; chiara.moltrasio@policlinico.mi.it (C.M.); davide.riva@unimi.it (D.R.); silvia.ferrucci@policlinico.mi.it (S.M.F.); angelo.marzano@unimi.it (A.V.M.); 2Department of Pathophysiology and Transplantation, Università degli Studi di Milano, 20122 Milan, Italy; 3Research & Innovation S.R.L. (R&I Genetics), 35127 Padova, Italy; colavito@researchinnovation.com; 4Pediatric Dermatology Unit, Department of Clinical Sciences and Community Health, Fondazione IRCCS Ca’ Granda Ospedale Maggiore Policlinico, 20122 Milan, Italy; gtadinicmce@unimi.it (G.T.); michela.brena@policlinico.mi.it (M.B.)

**Keywords:** atopic dermatitis, atopy, Netherton syndrome, frameshift mutation, genetic polymorphism

## Abstract

Netherton syndrome (NS) is a rare autosomal recessive disorder caused by *SPINK5* mutations, resulting in a deficiency in its processed protein LEKTI. It is clinically characterized by the triad of congenital ichthyosis, atopic diathesis, and hair shaft abnormalities. The *SPINK5* (NM_006846.4): c.1258A>G polymorphism (rs2303067) shows a significant association with atopy and atopic dermatitis (AD), which share several clinical features with NS. We describe an NS patient, initially misdiagnosed with severe AD, who carried the heterozygous frameshift (null) mutation (NM_006846.4): c.957_960dup combined with homozygous rs2303067 in the *SPINK5* gene. Histopathological examination confirmed the diagnosis, whereas an immunohistochemical study showed normal epidermal expression of LEKTI, despite the genetic findings. Our results corroborate the hypothesis that haploinsufficiency of *SPINK5*, in the presence of a *SPINK5* null heterozygous mutation in combination with homozygous *SPINK5* rs2303067 polymorphism, can be causative of an NS phenotype, impairing the function of LEKTI despite its normal expression. Due to the clinical overlap between NS and AD, we suggest performing *SPINK5* genetic testing to search for the *SPINK5* (NM_006846.4): c.1258A>G polymorphism (rs2303067) and ensure a correct diagnosis, mainly in doubtful cases.

## 1. Introduction

Netherton syndrome (NS) (OMIM #256500; ORPHA:634) is a rare autosomal recessive disorder, clinically characterized by the triad of congenital ichthyosis, hair shaft abnormalities, and atopic diathesis, with an estimated incidence ranging from 1:100,000 to 1:200,000 live births and a prevalence of 1–9/1,000,000 [1]. It is thought that the incidence may be higher (1:50,000), and this underestimation may be due to the difficulty in its diagnosis, as the symptoms resemble those of atopic dermatitis (AD), some forms of ichthyoses, and other conditions presenting with congenital erythroderma [2]

At birth, patients can present with erythroderma that could persist throughout life or evolve into polycyclic waxing and waning erythematous plaques with a characteristic double-edged scaling border, known as ichthyosis linearis circumflexa (ILC) [1]. Failure to thrive, hydroelectric disbalance, growth retardation, and recurrent infections are the most common complications during the infancy, whereas in adolescence and adult life, an eczematous-like and ILC phenotype is frequently observed, which is associated with atopic comorbidities and is often misdiagnosed for atopic dermatitis [1]. Trichoscopy represents a useful diagnostic tool for identifying trichorrhexis invaginata (IT), which is considered a hallmark of NS [3]. IT, also known as “bamboo hair”, is an abnormality of the hair shaft that causes it to invaginate at several points along the shaft; when the distal hair shaft intussuscepts, the proximal hair shaft creates a “ball and socket” appearance, whereas when the hair shaft ruptures at the site of this abnormality, the distal end of the remaining hair shaft shows a “golf tee-like” invagination [3]. 

The causative gene, serine peptidase inhibitor, Kazal type 5 (*SPINK5*), encodes the multidomain serine protease inhibitor LEKTI (lymphoepithelial Kazal-type related inhibitor), containing 15 potential inhibitory domains. The encoded preproprotein is proteolytically processed by a furin-driven proteolytic activation cascade to generate multiple protein products, which may exhibit unique activities and specificities; it plays a pivotal role in skin and hair morphogenesis, as well as in the anti-inflammatory and antimicrobial protection of mucous epithelia [4]. 

To date, more than 80 pathogenic variants have been described in NS patients [5], and most *SPINK5* mutations lead to the decay and/or the complete proteolytic breakdown of LEKTI, resulting in the persistent activation of skin kallikreins (KLK) and increased premature degradation of desmosomal and corneodesmosomal cadherins [6]. The dysfunctional epidermal homeostasis and the detachment of the stratum corneum give rise to the peculiar cutaneous phenotypes of this syndrome [1]. 

Defective cornification may also promote recurrent infections and inflammation in NS patients, mainly through the KLK5- protease-activated receptor 2 (PAR2) signaling pathway, and leads to the increased expression of pro-inflammatory mediators such as thymic stromal lymphopoietin (TSLP), tumor necrosis factor (TNF)-α, interleukin (IL)-8, and intercellular adhesion molecule 1 (ICAM-1), thus exacerbating the inflammatory loop [7]. Nonetheless, the tissue specificity of LEKTI also suggests its specific role in the trachea, sinosanal epithelium, and thymus; in the latter, the abnormal maturation of T lymphocytes, in NS patients, may alter the regulation of Th2 responsiveness to allergens, which determines the acute hypersensitivity response and IgE levels [6]; however, other possible mechanisms for LEKTI’s influence on IgE-mediated atopic responses have been proposed [8].

Currently, there are no satisfactory treatments for NS; different strategies, including antibacterial agents, emollients, and topical corticosteroids, are used to relieve the disease, whereas intravenous immunoglobulin and biologic agents (e.g., dupilumab) are used to attempt to treat NS patients with severe illness [9]. Furthermore, bacteriophage- and gene-based therapies may be an alternative option to control the symptoms and complications of NS syndrome, reducing disease severity and improving the patient’s quality of life [10]. 

Early confirmation of the diagnosis is therefore essential to appropriate patient management [11]; accurate clinical, histological, and dermoscopic characterization and genetic testing for *SPINK5* is necessary, especially in atypical cases where the classical features are not found. 

Herein, we report an NS patient, initially misdiagnosed with severe atopic dermatitis, who carried a heterozygous frameshift (null) mutation combined with a homozygous single-nucleotide polymorphism of the *SPINK5* gene.

## 2. Case Presentation

Following written informed consent, skin biopsy specimens were collected from the patient for histopathological/immunohistochemical characterization, and peripheral blood samples were obtained from the patient and her family members for genetic analysis at the Dermatology Unit of Fondazione IRCCS Ca’ Granda Ospedale Maggiore Policlinico, Milan, Italy. All procedures adhered to the principles of the Declaration of Helsinki.

A 31-year-old female patient first attended our department for a long-standing severe atopic dermatitis refractory to standard treatments. She had a history of eczematous skin eruption since infancy and multiple atopic comorbidities, including polysensitization to aeroallergens, dog epithelium, and cat dander; allergic rhinitis and conjunctivitis; allergic asthma (in remission at the time of the visit); and food allergies (nuts and milk).

Laboratory examination revealed an increased total immunoglobulin E (IgE) count, ranging from 200 to 1000 IU/mL (normal range < 120 IU/mL). She had been treated previously with topical and oral corticosteroids and cyclosporine with some clinical benefits, but frequent relapses occurred after the discontinuation of treatments. 

Clinical examination showed numerous erythematous and scaling lesion, most of which presented with an annular and polycyclic pattern and an incomplete double edge of peeling scale, especially on the arms and upper trunk, neck, and peri-areolar skin, associated with scratching lesions (Figure 1a–d). The patient experienced itch and stated that the lesions were migratory, arising after spontaneous remission in new body areas. 

Furthermore, she complained of dry, brittle, and difficult-to-comb hair since childhood; clinical and dermoscopic hair examination revealed features highly suggestive of trichorrhexis invaginata (Figure 2). 

Based on the anamnestic and clinical/dermoscopic picture, Netherton syndrome was suspected; histopathological/immunopathological examinations, as well as a genetic test, were carried out to confirm the diagnosis.

Her parents were healthy without any sign of skin disease, while her brother had suffered from atopy and atopic dermatitis since childhood. 

A 4 mm punch biopsy of lesional skin was obtained from the patient’s right arm; subsequently, sections were stained with hematoxylin and eosin, according to the standard protocol [12]. Histological examination revealed parakeratosis and subcorneal microabscesses with eosinophils, neutrophils, and extravasated blood cells. Dermal edema was present, accompanied by inflammatory infiltrates in the superficial dermis that mainly consisted of neutrophils (Figure 3a–c). 

LEKTI immunohistochemistry (IHC) was also performed to evaluate epidermal protein expression. IHC was carried out on the patient’s skin biopsy using a LEKTI polyclonal antibody (HPA009067) according to the manufacturers’ instructions. Modest positive staining in the upper spinous and granular layers of the epidermis was detected (Figure 3d).

EDTA-anticoagulated venous blood samples were collected from the proband and her family members to perform Next-Generation Sequencing (NGS) of the *SPINK5* gene. DNA extraction from peripheral blood samples was performed using a QIAamp DNA Mini kit (Qiagen, Valencia, CA, USA) according to the manufacturer’s instructions; subsequently, DNA concentration was measured using a Qubit 3.0 Fluorometer (Invitrogen, Life Technologies, Van Allen Way, Carlsbad, CA, USA). After DNA extraction and quality control, the regions of interest were enriched using DNA capture probes targeted against the exonic regions of *SPINK5*. The captured library was subsequently sequenced on an Illumina^®^ platform. A coverage depth of at least 50× was obtained for all targeted bases. Raw sequence data analysis was performed using designed software. Analysis of the entire coding region of *SPINK5*, including 10 bp of flanking intronic sequences and ~800 bp upstream of the first exon, was also performed. 

In the proband, a heterozygous frameshift variant in the *SPINK5* gene (chromosome 5q32, exon 11), NM_006846.4: c.957_960dup (p.Pro321fs), which gives rise to a premature stop codon, was identified. Based on guidelines from the American College of Medical Genetics and Genomics and the Association for Molecular Pathology (ACMG/AMP) [13], the c.957_960dup was predicted to be a likely pathogenic variant (PVS1_very strong + PM2_moderate), whose frequency is not reported in the gnomAD database. This heterozygous frameshift (null) mutation was accompanied by the presence of homozygous NM_006846.4 (*SPINK5*): c.1258A>G variant, which leads to the replacement of the amino acid glutamine with lysine in position 420 (p.Lys420Glu), located on domain 6 (exon 14) of *SPINK5* (Figure 4). This missense variant was rated benign, with a Combined Annotation-Dependent Depletion (CADD) score of 7.826 and a Genomic Evolutionary Rate Profiling (GERP) score of 2.84. Based on its frequency in the ExAC database [14], it has been reclassified as a polymorphism (rs2303067). However, rs2303067 (A; A genotype) has been significantly associated with susceptibility (1.8x risk) to atopic dermatitis [8] and asthma [15], and it has been formerly titled susceptibility to atopy, susceptibility to atopic dermatitis, and susceptibility to asthma.

Segregation studies on the clinically asymptomatic parents and atopic brother were performed through Sanger sequencing of the identified *SPINK5* variants.

In the patient’s mother, NM_006846.4 (*SPINK5)*: c.957_960dup was absent but NM_006846.4 (*SPINK5)*: c.1258A>G polymorphism was identified in a heterozygous carrier state. 

Her father carried both the NM_006846.4 *(SPINK5)*: c.957_960dup and c.1258A>G polymorphisms in a heterozygous state. 

The atopic brother only showed homozygous NM_006846.4 *(SPINK5)*: c.1258A>G polymorphism (rs2303067).

## 3. Discussion 

Netherton Syndrome (NS) is a rare autosomal recessive disease with skin, hair, and immune abnormalities, whose diagnosis is straightforward in the presence of characteristic dermatological features, trichorrhexis invaginate, and atopic diathesis [1,3]. However, it is still challenging to perform an accurate diagnosis in NS cases that lack a typical phenotype. Some diagnostic tests have facilitated the recognition of atypical cases through LEKTI skin immunodetection and mutation analysis of the *SPINK5* gene [11]. 

The *SPINK5* gene encodes for the serin protease inhibitor lymph-epithelial Kazal-type related inhibitor (LEKTI), a multifunctional protease inhibitor that is mainly expressed in the epidermis and controls the activity of stratum corneum trypsin- and chymotrypsin-like enzymes (SCTE/SCCE) [16]. Defective LEKTI function leads to the hyperactivation of SCTE/SCCE-like enzymes, causing reduced expression of desmoglein 1 (Dsg1) and desmocollin 1 (Dsc1), finally resulting in skin barrier impairment and altered keratinization [17]. LEKTI deficiency or loss can also lead to the typical morphological change of trichorrhexis invaginata through increased cross-linking in hair keratin structures and local defects of the inner root sheath [3]. 

The generation and analysis of knockout *Spink5*-/- mouse models [18,19], which mimick the key features of NS syndrome, confirmed that LEKTI deficiency leads to defective stratum corneum adhesion secondary to epidermal protease hyperactivity. Furthermore, Briot et al. [19] reported that unregulated KLK5 induces atopic dermatitis-like lesions through PAR2-mediated thymic stromal lymphopoietin expression in LEKTI-deficient epidermis, highlighting the pivotal role of protease signaling in skin inflammation and atopic responses. 

NS histological and ultrastructural analyses reveal incomplete keratinization and a broadly reduced, sometimes absent, granular layer. Parakeratosis, psoriasiform hyperplasia, subcorneum, or intracorneum splitting; microabscesses; and dermal inflammatory infiltrates consisting of neutrophils and/or eosinophils are often present in NS lesional skin biopsies and, particularly when seen in combination, can aid in performing a correct diagnosis [11]. 

In addition, immunofluorescence and immunohistochemistry show the absence of LEKTI expression in the granular and upper spinous layers in NS lesional skin, confirming the diagnosis in most cases [20,21].

Based on its tissue specificity, the loss of LEKTI in the trachea can lead to the destruction of the airway’s epithelial barrier, increasing sensitivity to inhaled allergens, whereas decreased expression of LEKTI in the sinonasal epithelium has been associated with allergic rhinitis [22]. Other potential mechanisms through which LEKTI can influence common atopic diseases, such as asthma, atopy, and atopic dermatitis, have been proposed: (i) proteinase-activated receptors are found in keratinocytes, and can act as targets for mast cell proteinases [23]; (ii) in the thymus, LEKTI seems to play a key role in T and B lymphocyte maturation and in antigen handling within other thymic cells [24]; and (iii) many allergens are serine proteinases, and this proteinase activity could be involved in the presence of different proteinase inhibitors [8]. In addition, loci influencing atopic diseases have been localized to the chromosome 5q32, where SPINK5 is located [8]. 

To date, more than 80 pathogenic variants have been described in NS patients, mainly including nonsense mutations and small nucleotide deletions/insertions, which alter the reading frame, thus producing premature termination codons, and subsequently, truncated proteins with loss-of-function. Splicing and missense mutations have been described, as well [5]. Most of these mutations are homozygous, and over 40% of them come from consanguineous families [5]. Compound heterozygosity, mainly in exon 16 of the *SPINK5* gene, is another, although less frequent, mechanism leading to LEKTI impairment, as reported in [5,25,26,27]. 

The heterogenous phenotype of NS mainly occurs due to the variable expressivity of *SPINK5* variants, and ranges from mild clinical signs to life-threatening complications, especially in the neonatal period, which may explain the efforts made to establish a solid genotype–phenotype correlation [28]. Overall, a more severe phenotype has been observed in some patients, with truncating variants located early in the coding sequence, leading to a deficiency or absence of LEKTI expression [5], whereas frameshift mutations near the C-terminal region might result in the retention of functional LEKTI fragments, leading to a milder phenotype [5]. Mutations affecting the splicing mechanism could be related to both a severe and a mild phenotype, compared to the proportion of normally spliced transcripts. Finally, some deep intronic pathogenic variants can activate the hidden splicing junction sites, allowing for a low-level production of the LEKTI protein, and thus, causing a milder phenotype [28]. Ten out of two hundred NS patients described in the literature are heterozygotes with variable disease severity, probably due to additional hidden pathogenic variants or mutations resulting in a gain of function, which is sufficient to cause the NS phenotype; however, this latter mechanism has never been reported [5]. Notably, some patients present a more severe phenotype than expected according to their genotype; this could be the result of aberrant translation products that escape the nonsense-mediated mRNA decay mechanism, thus acting in a dominant-negative manner and leading to a NS phenotype [5]. 

Intriguingly, Di et al. [29] reported the case of a patient with a heterozygous null mutation (2458insA in a noncoding region) combined with a c.1258A>G polymorphism (rs2303067) on both alleles of *SPINK5*; despite the normal expression of LEKTI mRNA and protein synthesis, this led to impaired protein function as well as the abnormal expression of skin barrier proteins, finally resulting in an NS phenotype consisting of ichthyosiform erythroderma, trichorrhexis invaginata, and hyperimmunoglobulin E. 

The single-nucleotide polymorphism c.1258A>G (rs2303067) in the *SPINK5* gene is a multi-allele variant of which the risk allele A is the most frequent, with a frequency of 48% as reported by the gnomAD database; it causes the replacement of glutamine with lysine in position 420 (p.Lys420Glu) in LEKTI, and has been associated with atopy and atopic dermatitis, correlating with skin barrier functioning, disease severity, and a higher risk of food allergy [8,30,31], highlighting its role in AD susceptibility [29]. 

Fortugno et al. [32] provided functional evidence that the homozygous p.Lys420Glu variant (rs2303067) impacts LEKTI function by increasing the furin-mediated cleavage rate of LEKTI precursors within the linker region D6–D7, thus preventing the formation of the LEKTI fragment D6D9, which is known to display the strongest inhibitory activity against SCTE-mediated Dsg1 degradation. Consequently, the authors showed increased activity of KLK5, KLK7, and eleastase-2, together with enhanced epidermal protease activity; the latter correlated with both reduced DSG1 protein expression and increased profilaggrin proteolysis. Increased expression of the proallergic cytokine TSLP was also reported. All these alterations, induced by the presence of the p.Lys420Glu variant within the LEKTI coding sequence, contribute to skin barrier permeability impairment, thus supporting the association studies that identified this variant as a predisposing factor to AD and atopy. 

All these findings suggest that the rs2303067 in *SPINK5* alone could impair LEKTI function and skin barrier integrity, and when combined with a pathogenic variant, might cause NS, due to the haploinsufficiency of *SPINK5*. 

Our case shares many similarities with the one presented by Di et al. [29]; our patient carried a heterozygous frameshift mutation (NM_006846.4 (*SPINK5*): c.957_960dup; p.Pro321fs), previously reported in a homozygous state in an Italian patient [33], and a c.1258A>G polymorphism on both alleles of *SPINK5*, thus confirming that the haploinsufficiency of *SPINK5*, due to the presence of one null mutation with homozygous c.1258A>G polymorphism, could be causative of the NS phenotype with impaired LETKI function, despite its normal mRNA expression.

In different terms, *SPINK5* mutation in one allele combined with a c.1258A>G polymorphism on the other allele led to the normal expression of both LEKTI mRNA and protein synthesis, but with insufficient function, due to the polymorphism, to maintain its normal biological role, thus causing an NS phenotype [29].

Despite the same pathomechanism, our patient presented an ichthyosis linearis circumflexa phenotype, whereas the patient described by Di et al. revealed ichthyosiform erythroderma and an absence of eczematous lesions [29]; this result could be due to the different null heterozygous mutations involved in this pathogenetic scenario. 

To date, this is the second case described in the literature on NS occurring due to the haploinsufficiency of the *SPINK5* gene, in the presence of one null mutation with homozygous c.1258A>G polymorphism, acting as a “true” mutation.

## 4. Conclusions

In conclusion, although clinical, trichoscopical, and histopathological features represent useful tools for NS diagnosis, due to the possibility of misdiagnosis with a severe form of AD (especially in adults), we suggest performing *SPINK5* genetic testing; searching for a c.1258A>G polymorphism (rs2303067), especially in doubtful cases, to ensure a correct diagnosis; and genetic counselling for the patients and their relatives.

Moreover, NS patients have a heterogenous phenotype, mainly due to the variable expressivity of *SPINK5* variants; genetic characterization of each NS patient is thus essential to establish a robust genotype–phenotype correlation, to increase knowledge of the pathogenetic mechanisms underlying this disease, and to drive targeted molecular therapy. Further research, including cellular, molecular, and functional approaches, is warranted to define the functional impact of each variant on LEKTI and unravel the exact pathomechanisms underlying NS, in both typical and atypical cases. 

## Figures and Tables

**Figure 1 genes-14-01080-f001:**
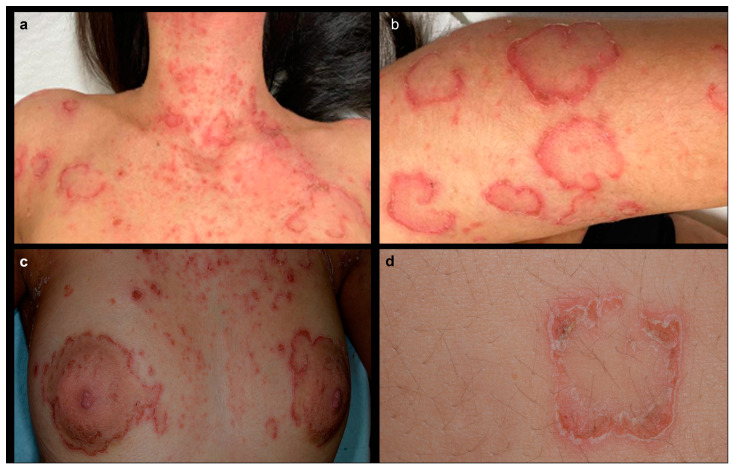
Clinical findings of the patient showing (**a**) annular erythematous and scaling plaques mixed with eczematous lesions on the upper trunk and neck; (**b**) annular and circinate erythematous lesion with a peeling scale on the left arm; (**c**) a detail of the same annular erythematous and scaling lesion surrounding peri-areolar skin; (**d**) a detail of an annular lesion showing a double edge of peeling scale.

**Figure 2 genes-14-01080-f002:**
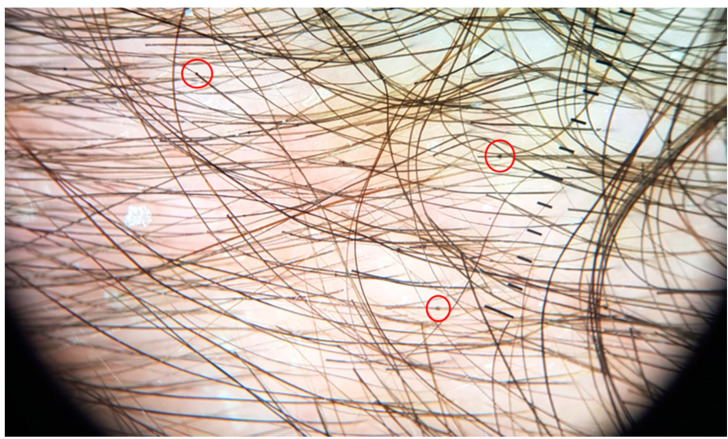
Trichoscopic image of the patient showing dark nodular structures clearly visible on some of the hair shafts (highlighted in red circles), resulting in a “golf tee” appearance of the hairs, a characteristic feature of trichorrhexis invaginata.

**Figure 3 genes-14-01080-f003:**
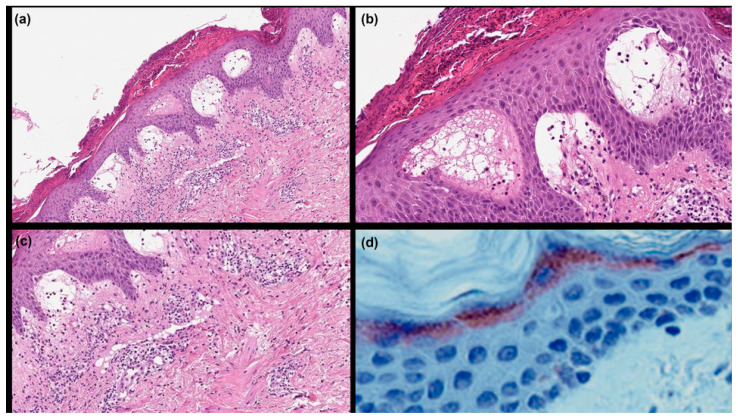
Histopathology showing (**a**) parakeratosis and subcorneal acantholysis forming microabscesses with dermal edema, accompanied by inflammatory infiltrate in the superficial dermis (hematoxylin and eosin staining, ×20); (**b**) close-up view revealing parakeratosis and epidermal acantholysis with vesicles filled with eosinophils, neutrophils, and extravasated blood cells (hematoxylin and eosin staining, ×40); (**c**) close-up view showing a dermal edema with an inflammatory infiltrate, mainly consisting of neutrophils (sometimes extending into the epidermal ridges) and eosinophils with scattered macrophages and lymphocytes (hematoxylin and eosin staining, ×20). Immunohistochemistry showing (**d**) modest positive staining for LETKI in the upper spinous and granular layers of epidermis (LEKTI polyclonal antibody [HPA009067] ×100).

**Figure 4 genes-14-01080-f004:**
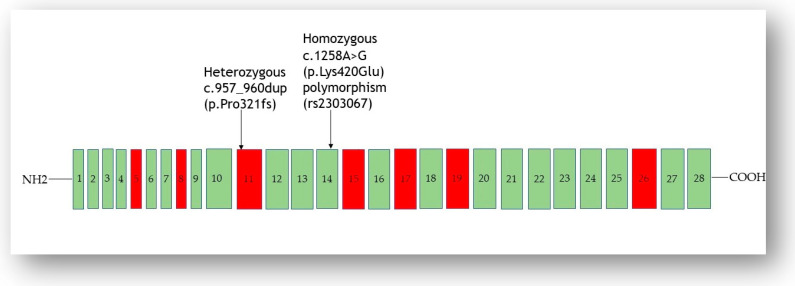
Mutations of the *SPINK5* gene occurring in our patient: a heterozygous duplication (exon 11) combined to a homozygous polymorphism (rs2303067) in exon 14. Typical exon domains are represented in red and other domains in green. Exons 29–33 are not shown because no mutation has been detected, to date, in NS patients.

## Data Availability

The data are contained within the article.

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
