# Peer review of "Netherton Syndrome Caused by Heterozygous Frameshift Mutation Combined with Homozygous c.1258A>G Polymorphism in SPINK5 Gene"

_genes, 2023, doi:10.3390/genes14051080_

Round 1

Reviewer 1 Report

The authors report a patient with Netherton syndrome, in whom they identified a heterozygous pathogenic variant and a homozygous missense SNP of the SPINK5 gene. The brother of the affected patient carries the homozygous missense SNP only and affected by atopic dermatitis. The parents are clinically unaffected and the mother carriers the pathogenic variant in heterozygous form. 

The authors concluded that the combination of the homozygous missense SNP and the heterozygous pathogenic variant is responsible for the development of Netherton syndrome.

Suggestions:

The authors numbered the identified variants according to NM_001127698.2 and thus it is not possible to check them in the Ensemble genome browser or in the Franklin variant effect analyser.

1. Please provide the precise numbering of the identified variants according to NM_006846.4 or add the pictures of the sequenograms of the variants.

2. Regarding the identified SNP: Please add the rs number of the identified SNP. Please add information whether it has already been published in Netherton syndrome and whether there is any functional data about it supporting its suggested pathogenic role. Does it affect any known functional domain of the protein? Does it located in an evolutionary conserved region of the protein? Does it located in an enhancer region? Please add its population genetic data including MAF. Please add the predicted consequences of the SNP.

3. Please add CADD and GERP data of both variants.

4. Please consider to use MLPA for the detection of a deletion or a duplication. The heterozygous pathogenic variant is one of the variant contributing for the development of Netherthon syndrome in your patient. The question is whether really the homozygous SNP is the other disease-causing variant. Please consider to use SPINK5 MLPA for the detection of a deletion or a duplication.

5. The authors used WES, please check WES data for CNVs.

6. Is missing heritability is an existing phenomenone in Netherton syndrome? In how many % of the patients, could genetic investigations elucidate the genetic background of the disease? What is the proportion of the patients with one heterozygous pathogenic variants?  

Reviewer 2 Report

The authors have presented an interesting case report of NS with heterozygous frameshift mutation combined with homozygous c.1258A>G polymorphism in SPINK5 gene .  The overall scope is well met. The clinical, trichoscopy and HP pictures are in good quality. The introduction is concise and informative. However, there are a few concerns that can be appropriately addressed to improve the paper. My remarks are the following: 

  1. The article need english/ style correction (e.g. line 202)
  2. The discussion could be improved (e.g. comparison with more cases).

The authors have presented an interesting case report of NS with heterozygous frameshift mutation combined with homozygous c.1258A>G polymorphism in SPINK5 gene .  The overall scope is well met. The clinical, trichoscopy and HP pictures are in good quality. The introduction is concise and informative. However, there are a few concerns that can be appropriately addressed to improve the paper. My remarks are the following: 

  1. The article need english/ style correction (e.g. line 202)
  2. The discussion could be improved (e.g. comparison with more cases).
